# What is the Co-Creation of New Knowledge? A Content Analysis and Proposed Definition for Health Interventions

**DOI:** 10.3390/ijerph17072229

**Published:** 2020-03-26

**Authors:** Tania Pearce, Myfanwy Maple, Anthony Shakeshaft, Sarah Wayland, Kathy McKay

**Affiliations:** 1School of Health, University of New England, Armidale, NSW 2351, Australia; mmaple2@une.edu.au; 2National Drug and Alcohol Research Centre, University of New South Wales, Randwick Campus, 22–32 King Street, Randwick, NSW 2031, Australia; a.shakeshaft@unsw.edu.au; 3C43A, Jeffrey Miller Admin Building, Cumberland Campus, The University of Sydney, Lidcombe, NSW 2141, Australia; sarah.wayland@sydney.edu.au; 4Department of Health Services Research, University of Liverpool, Liverpool L69 3BX, UK; kmckay@tavi-port.nhs.uk; 5Tavistock and Portman NHS Foundation Trust, University of Liverpool, Liverpool L69 3BX, UK

**Keywords:** co-creation, co-creation of new knowledge, content analysis, knowledge translation, collaboration, health intervention, knowledge production, co-design, co-evaluation, co-ideation, co-implementation

## Abstract

Co-creation of new knowledge has the potential to speed up the discovery and application of new knowledge into practice. However, the progress of co-creation is hindered by a lack of definitional clarity and inconsistent use of terminology. The aim of this paper is to propose a new standardised definition of co-creation of new knowledge for health interventions based on the existing co-creation literature. The authors completed a systematic search of electronic databases and Google Scholar using 10 of the most frequently used co-creation-related keywords to identify relevant studies. Qualitative content analysis was performed, and two reviewers independently tested the categorisation of papers. Of the 6571 papers retrieved, 42 papers met the inclusion criteria. Examination of the current literature on co-creation demonstrated how the variability of co-creation-related terms can be reduced to four collaborative processes: co-ideation, co-design, co-implementation and co-evaluation. Based on these four processes, a new definition of co-creation of new knowledge for health interventions is proposed. The analysis revealed the need to address the conceptual ambiguity of the definition of “co-creation of new knowledge”. The proposed new definition may help to resolve the current definitional issues relating to co-creation, allowing researchers and policymakers to progress the development of co-creation of new knowledge in research and practice.

## 1. Introduction

Researchers, practitioners and policy makers have a strong interest in increasing the speed and efficiency with which research findings contribute to improved public health outcomes. The most frequently cited translational models that facilitate research findings into practice are: RE-AIM (Reach, Effectiveness, Adoption, Implementation, and Maintenance); “T” models (Translation Research Continuum); and KTA (Knowledge to Action) [1]. RE-AIM is an evaluation framework that measures the impact of health interventions on improvements to public health outcomes [1]. The “T” model framework follows a six-stage linear process where research moves from a discovery or “basic” stage (T0) through to the uptake of research findings into clinical or public health practice (T5) [2,3]. The KTA takes a systems approach where new knowledge is created and applied into practice through two interconnected processes: knowledge creation and action research [1]. Although these translational models have been widely applied, there is no empirical evidence on whether their application actually improves the uptake of research findings into practice. There is also a paucity of empirical evidence on how these translational models, for example those used in health interventions, are interpreted or used by different stakeholders at different points in the processes that they describe [1]. Beyond translational models, Community-Based Participatory Research [4] or Action Research [5] are alternate practices involving multiple stakeholders; however, there is generally no requirement for a collaborative commitment throughout the whole process from the design phase, and there are often power dynamics that remain unresolved. The result can be that the researchers drive the initial concepts and then leave post intervention trial, taking the knowledge gained with them [6], thus limiting ongoing benefit to those for whom the intervention was being designed.

Despite conceptual appeal, translational models may be of limited practical benefit for a number of reasons. First, a pre-condition for translating research into practice is that research findings should be readily available, of good methodological quality, and provide useful and useable evidence for those working at multiple levels (community to policy). It has been identified that a substantial proportion (40%–89%) of published research does not meet these criteria: (1) the papers did not include sufficient detail to allow their results to be useful or replicable; (2) they did not take into account existing evidence on the same questions; or, (3) they contained readily avoidable design flaws [7]. Second, numerous systematic reviews across a wide range of content areas have identified a minority of studies (ranging from an estimated 5% to 25%) report on methodologically sound evaluations of interventions aimed at identifying best evidence-based practice, meaning there is relatively little published research readily able to be translated into improved services or policies [8,9,10,11,12,13]. Third, there is traditionally minimal collaboration between academics, service providers, communities and policymakers in determining the most important research questions and the most appropriate evaluation methods, meaning published research findings are often of limited practical value. Fourth, despite the methodological benefits of well-controlled evaluation designs, such as Randomised Controlled Trials (RCTs), and their perceived desirability among researchers and academic journal editors, typically the evidence they generate has high internal validity and low external validity (generalisation), meaning the practical applicability of the results of such highly controlled trials is usually variable [14,15]. Fifth, dissemination strategies often fail to convey the importance of research evidence that has established the benefits and costs of interventions, which limits the demand for using translational models to transfer that evidence into practice [16].

Given these limitations, there is scope to develop additional approaches to improve the speed and efficiency with which research findings contribute to improved health interventions and outcomes. These alternatives do not need to replace existing translational models but would complement them. The common principle across current translational models is a focus on reducing the time gap between discovering new knowledge through research and the uptake of that new knowledge into practice (that is, into the delivery of services, programs or treatments or its integration into policy). A complementary framework, which conceptualises the generation of new knowledge as occurring alongside the delivery of health interventions in organisations, addresses the aforementioned limitations. An example of how this framework could be applied in practice includes mental health organisations involved in the delivery of health interventions such as suicide prevention programs. Health interventions are defined as those interventions creating change in services, treatment or policies, resulting in better health outcomes [17]. In this scenario, the service providers and researchers collaborate to embed the collection of data (research evaluation) into the routine delivery of services. The collection of data therefore occurs alongside the delivery of the health intervention targeting suicide prevention. The framework would then rely on identified parties co-creating the evidence and the outcome and, thus, the knowledge obtained.

Some research has focused on “how to” co-create, especially in health and community settings [18]; however, there remains a lack of consensus on the meaning and use of the term co-creation of new knowledge. Many terms are used interchangeably and with ill-defined or no definition as to the meaning behind the terms. A review of the existing literature showed co-creation (also referred to as co-design and co-production) is conceptualised and operationalised in many different ways even within the same field. In health, for instance, the current trend is to depict co-creation as a model of participatory research [19,20]. Others define co-creation as the fusion of two concepts (community-based participatory research and integrated knowledge translation [21], while some researchers [22,23] base their understanding on a model devised by Sanders and Stappers. In the latter example, co-design is described as a collection of activities ranging from ideation to planning and evaluation. Despite the lack of consensus, two specific definitions of co-creation have been proposed to resolve some of this conceptual ambiguity: (1) “a process whereby researchers and stakeholders jointly contribute to the ideation, planning, implementation and evaluation of new services and systems *as a possible means to optimise the impact of research findings*” [22]; and, (2) “*the collaborative generation of knowledge* by academics working alongside stakeholders from other sectors” [20]. Although both definitions share the concept of equitable collaboration between stakeholders, neither definition appears to adequately capture the concept of co-creation as simultaneously focusing on both program or policy delivery and the generation of new knowledge. The first definition focuses on the former (see italicised text), and the second focuses on the latter (see italicised text).

The lack of a universally accepted definition creates unnecessary ambiguity [24,25,26,27]. Researchers are not able to effectively search electronic databases and retrieve relevant studies, which inhibits the development of a coherent, critical mass of adequately homogenous co-creation research. It then becomes far more difficult for service providers and policymakers to engage in co-creation activities because they are being asked to engage in a process that either lacks clarity or is highly variable across different researchers and disciplines. A number of factors are likely to be maintaining the current conceptual ambiguity of co-creation. For example, ‘co-creation’ is used widely in different fields of practice, such as business management, technology, tourism, marketing and, more recently, health. This has generated a range of specific applications of the concept that, at least on face value, may have implied that it is a different concept applied in different contexts, rather than the same concept adapted to different contexts [28]. As a result, these perceived conceptual differences are exacerbated by different fields of practice attributing different levels of importance to the component processes within co-creation, such as co-ideation and co-evaluation. It must also be remembered the concept of the co-creation of new knowledge is at a relatively early stage of evolution, which means it will initially be characterised by a diverse set of co-created related terms which will solidify into more standardised and accepted lexicon over time [29].

With recent attention to these collaborative practices and scholars calling for a consensus on the use and meaning of co-related terms [30], the objective of this paper is to act as a starting point for debate and discussion on standardising the concept of co-creation of new knowledge. The aim of this paper is to propose a definition that is likely to have utility for those working in health interventions to standardize language to better inform others of the processes used. To achieve our aim, three steps were undertaken. First, the identification of contemporary studies that use a co-creation-related term. Second, the use of qualitative content analysis to assess the use of co-creation-related terms and examine patterns in their manifest attributes and meanings. Third, the use of the results of the data analysis to form a foundation for a new proposed definition of co-creation of new knowledge.

## 2. Materials and Methods

We conducted a qualitative content analysis of existing definitions and/or descriptions of any collaborative activities to formulate a standardised definition of “co-creation of knowledge”. Content analysis is a method used for analysing information and interpreting its meaning using a systematic coding approach to identify trends, patterns and relationships in data [31]. It allows researchers to reliably perform an inductive analysis through systematic examination and constant comparative evaluation of meanings and context [32]. In using this approach, the authors followed the three phases of data analysis described by Elo and Kyngäs [33]: (1) preparation (unit of analysis and data collection); (2) organisation (coding and abstraction); and (3) reporting (synthesis of results), detailed below.

### 2.1. Preparation Phase

#### 2.1.1. Unit of Analysis and Data Collection Method

In preparation for the collection of data, published papers containing any description and/or definitions of collaborative activities (e.g., co-design, co-production etc.) were chosen as the unit of analysis [34]. The method of data collection involved searching for papers containing original or secondary definitions or descriptions of co-creation-related terms. As attention to these practices has grown considerably in recent years [35] and to ensure the literature we were obtaining is contemporary, we used a five-year range in our search. Each step is clearly described below. 

#### 2.1.2. Sampling Strategy

In the absence of a standardised list of co-creation-related keywords and/or established index headings (e.g., Medical Subject Headings (MeSH)) to identify papers, author (TP) performed a snowball search of Google and Google Scholar to compile a list of terms. Initially, the Google search focused on two keywords commonly appearing in the literature: “co-creation” and “co-production”. As this paper focused on the specific use of co-creation terminology, broader concepts such as “collaboration”, “mode 2”, “participatory action research” and “partnership” were excluded from the snowball search. No limits were placed on either publication date or publication type. This helped ensure the maximum number of differing terms were being identified. Scanning the titles and abstracts of the records retrieved resulted in an exhaustive list of co-creation-related synonyms. This process identified 22 unique keywords (Table 1).

#### 2.1.3. Constructing a Controlled List of Search Terms

The list of 22 keywords was reduced to those used most frequently, given the high probability that less frequently used terms, such as “co-development”, would be used simultaneously with more popular terms, such as “co-creation” and “co-production”. Using advanced search methods in two electronic databases—PubMed (Medline) and ProQuest (multidisciplinary databases)—the 22 keywords were ranked by frequency. As shown in Table 1, this process identified the 10 most frequently used keywords (>50 records retrieved).

#### 2.1.4. Search Protocol and Screening of Records

As shown in Figure 1, the 10 most frequently mentioned keywords were used to identify and review potentially relevant papers in a Preferred Reporting Items for Systematic Reviews and Meta-Analyses (PRISMA) compliant search [36]. Seven electronic academic databases were searched (Emerald, EBSCO (including CINAHL), Informit, ProQuest, PubMed, Scopus and Web of Science), as was Google Scholar to capture grey literature. The eligibility criteria included: (1) papers containing clear definitions or descriptions of co-creation-related processes or activities; (2) papers focused on the delivery of services, programs, policies or products; and (3) empirical and non-empirical papers published in either peer reviewed or grey literature. Both English and non-English citations (with English abstracts) were eligible. As earlier test searches of co-creation-related keywords with no date limitations retrieved a large amount of irrelevant material, the date range was limited to the period from 1 January 2014 to 1 November 2018. This time period allowed the retrieval of a representative sample of differing definitions and descriptions of contemporary co-creation-related terms. Author (TP) completed the search on the 3 November 2018. Search results were imported into Endnote X8, and duplicate citations were removed using the Endnotes’ duplicate identification tool. Rigorous manual checks for any remaining duplicates were also undertaken. The literature search, as shown in Figure 1, resulted in the identification of 12,094 articles, of which 5523 were duplicates, leaving 6571 papers for review. After exclusion of 6197 papers, the full-text versions of the eligible papers (n = 374) were exported from Endnote X8 into NVivo 11 Pro QSR. Author (TP) systematically checked and re-checked the 374 papers for evidence of definitions or descriptions of co-creation-related activities. At this stage, no detailed assessment of the definition was made, rather any identification of a process description resulted in the papers being included. Of these, 42 papers containing clear definitions or descriptions of co-created activities were identified. It is important to note that 69 additional papers also contained descriptions of co-created processes; however, the descriptions in these papers were too ambiguous to allow them to be categorised. For instance, some papers on co-design did not offer a clear and complete description of the co-design process. Instead, the authors of those papers focused on detailed reporting on the outcomes of the study while only providing a general descriptive overview of the level of involvement by the participants engaged in the process of co-design [18,37,38]. This lack of clarity around the co-design process and what was involved and the level of contribution by participants made it difficult for us to accurately categorise these papers.

All papers including manifest process descriptions met the eligibility criteria for full assessment, as demonstrated in Figure 1. 

### 2.2. Organization Phase

As only 42 papers met the inclusion criteria, author (TP) performed a manual process of inductive analysis. First, the data were analysed by the lead author through open coding (careful reading, highlighting key phrases and segments of text relating to descriptions or definitions of co-creation-related activities). Second, during the abstraction process, a coding scheme of four subcategories was formed through comparison of similar descriptions and meanings of co-creation-related terms. From this process, regardless of the terminology used in the papers, descriptions of co-related activities were aligned with one of four categories: “*co-ideation*”, “*co-design*”, “*co-implementation*” and “*co-evaluation*”. For example, if a paper referred to a process as “co-production” but was actually describing an activity involving design, then it was classed as “co-design”. If a paper on “co-production” described a mixture of collaborative processes (e.g., co-implementation and co-evaluation), then the paper was categorised under both processes. 

## 3. Reporting and Results Phase

### 3.1. Variability of Terms and Description of Co-Creation Activities

The iterative comparison of 42 papers demonstrated the wide variability of co-creation-related terms being used in the literature. For instance, co-ideation was referred to using nine terms, with seven terms used to refer to each of the categories co-design, co-implementation and co-evaluation. In all, the literature used 18 unique terms to describe the four activities of co-ideation, co-design, co-implementation and co-evaluation. Of the 92 mentions of a co-creation-related activity identified in the 42 included papers, most terms were related to co-design and co-ideation (n = 36 and n = 26 respectively), followed by co-implementation (n = 17). Synonyms used to describe activities relating to co-evaluation were the least used terms (n = 12). The use of variable terms is evidence of a lack of standardisation in the use of co-creation-related terms across the four identified industries within which this concept appears. Business and marketing used co-creation-related terms most frequently (n = 28), followed by health and welfare sectors (n = 23), community-based (n = 21) and public policy (n = 20) sectors. 

### 3.2. Trustworthiness

The trustworthiness of the study was established using Lincoln and Guba’s (1985) [39] four criteria: credibility, dependability, confirmability and transferability. Credibility was reached through the use of the constant comparison method to ensure consistency in the categorisation of data and critical peer debriefing with co-researchers [40]. Dependability was established by having clear documentation of the data collection process and development of the coding frame and the use of correlation coefficient (ICC) where two co-authors (SW and KM) blind to the literature on co-creation categorised a random selection of 40% of the included papers into one or more of the four categories identified in Figure 1. Reliability between coders was calculated in Statistical Package for the Social Sciences (IBM SPSS 25) (IBM, Armonk, NY, USA), with coding scores demonstrating a good result (intraclass correlation coefficient 0.726; 95% confidence interval 0.624–0.810), as did the interrater reliability of the examination component (intraclass correlation coefficient 0.888; 95% confidence interval. 0.833–0.927). Confirmability was achieved through feedback from two of the co-authors who have experience as community practitioners, while transferability was achieved by searching and including literature from a broad cross section of disciplines.

### 3.3. Results

The results of our findings are presented in the following 3 tables:

Table 2 summarises the range of co-creation terms used in differing fields of practice appearing in papers published between 2014 and 2018. The identified terms and their descriptions were assessed and categorised according to four primary collaborative processes. In Table 3, examples of the key phrases and similar descriptions and meanings used by authors to define and describe attributes of co-creation are included. These attributes and meanings assigned by researchers and practitioners were then extracted during the organisational phase and used to formulate the operational definitions shown in Table 4.

## 4. Discussion

This study identified 42 papers published between 2014 and 2018 that provided a manifest definition and/or description of co-creation-related terms. Among these 42 papers, a co-creation-related term was mentioned 92 times. The range of terms varied widely: for example, co-ideation was described using nine terms. These 92 appearances of a co-creation-related term were readily collapsed into the 4 processes proposed in the standardised definition of the co-creation of new knowledge: co-ideation (26 times); co-design (36 times); co-implementation (17 times); and co-evaluation (12 times). Blinded coders (SW and KM) replicated the classification process, achieving good agreement between themselves and the first author in the classification of studies with a clear definition and/or description of co-creation. During the coding trial, there was considerable variation between coders when assessing papers with latent definitions. This event prevented the coding of papers with ambiguous definitions or descriptions of co-creation-related activities. It also reinforced the importance of establishing unambiguous definitions to optimise the consistent application of the concept of co-creation of new knowledge regardless of the user (researchers, service providers and public health policy practitioners).

Given the current variability and the potential to improve these existing definitions, this paper proposes a standardised definition for the co-creation of new knowledge based on the inductive analysis of the existing literature and input from co-researchers as community practitioners. Specifically, through this process using the content analysis model proposed by Elo and Kyngäs [33], we have achieved our aim of defining co-creation of new knowledge as:

The generation of *new knowledge* that is derived from the application of rigorous research methods that are *embedded* into the delivery of a program or policy (by researchers and a range of actors including service providers, service users, community organisations and policymakers) through *four collaborative processes*: (1) generating an idea (co-ideation); (2) designing the program or policy and the research methods (co-design); (3) implementing the program or policy according to the agreed research methods (co-implementation), and (4) the collection, analysis and interpretation of data (co-evaluation).

This definition comprises three core principles (indicated by the italicised text in the definition). Principle 1: new knowledge derives from the application of rigorous research methods. While specifying that new knowledge must derive from rigorous research methods may be tautological, these concepts are used separately to emphasise that co-creation of new knowledge is an under-utilised way of applying accepted scientific methods, not an alternative to them. This means commonly used frameworks, such as continuous quality improvement or participatory action, would only achieve co-creation of new knowledge if the methods used were sufficiently rigorous [88]. Principle 2: research methods are embedded into the delivery of a program or policy as a way to ensure the new knowledge has an immediate practical application, such as quantifying the impact of a program or policy, or the economic efficiency with which it is delivered. Principle 3: as summarised in Table 2, co-creation comprises four collaborative processes. Common across all of the included papers was the use of the prefix ‘co-’ representing collaboration and mutual engagement. Within each collaborative process, the level of participation and partnership between researchers and service providers may vary depending on the activity being undertaken [89] and the way the data are being collected. The new knowledge, however, would only be defined as being co-created if it comprised all four processes. Evidence suggests having input from all stakeholders across the entire co-creation process will result in stronger partnerships and a greater commitment by all stakeholders to use the knowledge produced [90].

### 4.1. Implications for Service Delivery and Policy Implementation

For service delivery and policy implementation, the benefits of using a standardised definition for the co-creation of new knowledge are threefold. First, it will allow service providers, policymakers and researchers to more easily differentiate between what is co-created knowledge and what is not. Currently, as shown in Table 2, the literature on co-creation is heterogeneous, and co-creation-related terms are applied without any clear consistency in their meaning. Second, improved clarity, both in the definition of co-creation of new knowledge and in key stakeholders’ understanding about it, is a necessary (although insufficient) step in facilitating a more frequent evaluation of programs and policies that will provide governments, funders and services with more immediate, more relevant and more high-quality evidence about which policies and programs are most effective and are good value for money. This contrasts with the current focus on translational models for utilising research evidence in practice which, as argued in the introduction, are of limited practical benefit to service providers and policymakers. Third, greater clarity about co-creation as a concept and an approach will assist in developing new and innovative ways of embedding research into practice because the processes required for embedded research are clearly specified. The new, applied and timely research evidence generated by greater use of co-creation processes will, in turn, build sustainability in the delivery of cost-effective programs and policies. Good quality evidence provides an unambiguous, transparent rationale that can be used to defend the provision of programs and policies when their existence is challenged by threats, such as funding cuts and organisational restructures. More frequent embedding of research into practice is also likely to encourage a greater focus from all stakeholders on improving outcomes for clients and target populations using rigorous and appropriate methods of data collection [91].

### 4.2. Implications for Future Research

There are four key ways in which the concept of co-creation of new knowledge can be developed. First, there is a need to develop a measure of co-creation of new knowledge (based on the definition) to capture the extent to which studies that claim to use a co-creation approach actually do so. The psychometric properties of such a measure would need to be established, including inter-and intra-rater reliability and validity (including content, construct and face validity). Similarly to the development of a measure for the extent to which co-creation is used in relevant papers, a measure of the extent and quality of collaboration between researchers and practitioners would be useful, given the three principles for co-creation proposed by Greenhalgh [20] emphasise the centrality of collaboration in co-creation of new knowledge. This concept has been applied elsewhere, such as in Pretty’s participation typology used by researchers to assess levels of community participation ranging from no participation to self-mobilisation [91,92]. The extent to which existing measures might be applicable to the co-creation of new knowledge, however, is unknown. Establishing a new co-creation measure will be important where evaluations suggest a program or policy is ineffective, because it would help clarify whether the apparent lack of effectiveness is a consequence of the program or policy itself, of inadequate application of the co-creation process (using a measure of co-creation) or of an under-developed partnership between the key stakeholders (using a measure of participation). Second, identifying when it is appropriate to use a co-creation process is important because these processes will not be applicable to all types of research [93]. As a general principle, co-creation processes are likely to be most well aligned with research that seeks to produce actionable or usable knowledge [93]. Third, adaptation of high-quality evaluation designs and measures that could be used in the co-creation of new knowledge would usefully allow for the lack of strict controls in service delivery. Service delivery providers exist in unstable environments, with a changing client base and funding pressures. The co-existence of researchers and service providers calls for evaluation designs that are adaptable to the needs of all stakeholders that are typically able to be achieved in the context of the routine delivery of services or the implementation of public policy [88]. Fourth, given researchers have very different key performance indicators (KPIs) than service providers and policymakers, establishing common KPIs, such as demonstrating the benefits and costs of programs or policies as they are implemented and using standardised co-creation of new knowledge processes, would encourage greater collaboration and strengthen the focus on outcomes. Further, maximising the value of co-creation of new knowledge will come from understanding the perspectives of the end-user (consumers, citizens, patients, governments, service providers and philanthropists) on the feasibility of co-creation to achieve social policy objectives and funding goals. Standardised terminology will also assist in future theory development and testing where these processes are used and clearly defined.

### 4.3. Strengths and Limitations

The study has four key limitations. First, this paper examined nearly five years of published co-creation literature. Limiting the search for papers by publication date was based on evidence that the current definitions and processes would be informed by earlier research findings. Furthermore, as the searches were conducted using multiple electronic databases, covering a broad spectrum of disciplines, the risk of bias to a specific discipline was reduced. Second, the outcome of the intraclass correlation co-efficient may have been compromised by the small sample size, as a number of samples above 30 is recommended [94]. The third limitation is that publications may have been misclassified, although the strength of agreement between coders in categorising the manifest papers suggests that this is unlikely. Fourth, the independent review of 40% of papers may be insufficient to establish that the definition of ‘co-creation of new knowledge’ can be applied consistently in the field. The adequacy of the proposed definition is based on a combination of descriptions of previously applied research processes and the knowledge and experience of field practitioners. The test conducted by the independent reviewers demonstrated general consensus with good agreement. A useful next step for research would be to explore with stakeholders and policymakers this issue in real time or, prospectively, to understand what they think co-creation might be defined as and then apply this to the existing published research. The findings of this paper have already been shared with organisations involved in co-creation activities, namely, those from the field of mental health and suicide prevention.

## 5. Conclusions

Although co-creation of new knowledge is presented as an alternative model for translating research, its use in industry is hindered by its conceptual immaturity. Evidence of a lack of definitional consistency is seen in the wide variability of terms used by industry professionals to describe co-creation. It is important for practitioners to understand such variability exists, as this could prevent double-work or excess use of limited resources when developing new community-based and targeted health initiatives. In this paper, a new standardised co-creation definition has been proposed, which has been developed from the existing activities and processes identified in the contemporary literature. This new definition will help to address the lack of clarity, initiate debate around building an evidence base on co-creation and demonstrate how the definition can be consistently applied. The practical novelty of this theoretical work is clear, as it allows practitioners and other healthcare workers and researchers to start with the same understandings and strategies when developing new healthcare interventions, making such development clearer and more straightforward. Also, by including in the definition the key principle of embedding research methods in the delivery of services may help to ensure a greater investment by practitioners in the research process and its outcomes. Advancement of co-creation of new knowledge as a concept will depend upon the future development of measures of co-creation to ensure its reliability and validity and the alignment of common key performance indicators to encourage greater collaboration between stakeholders. Future collaborations between researchers, service providers and consumers, building targeted health intervention using the four processes identified in the proposed model of co-creation of new knowledge, will likely reduce the timeframe between development of new interventions and community benefit. Using the three core principles proposed will clarify commitment and roles of all players in any health intervention developments.

## Figures and Tables

**Figure 1 ijerph-17-02229-f001:**
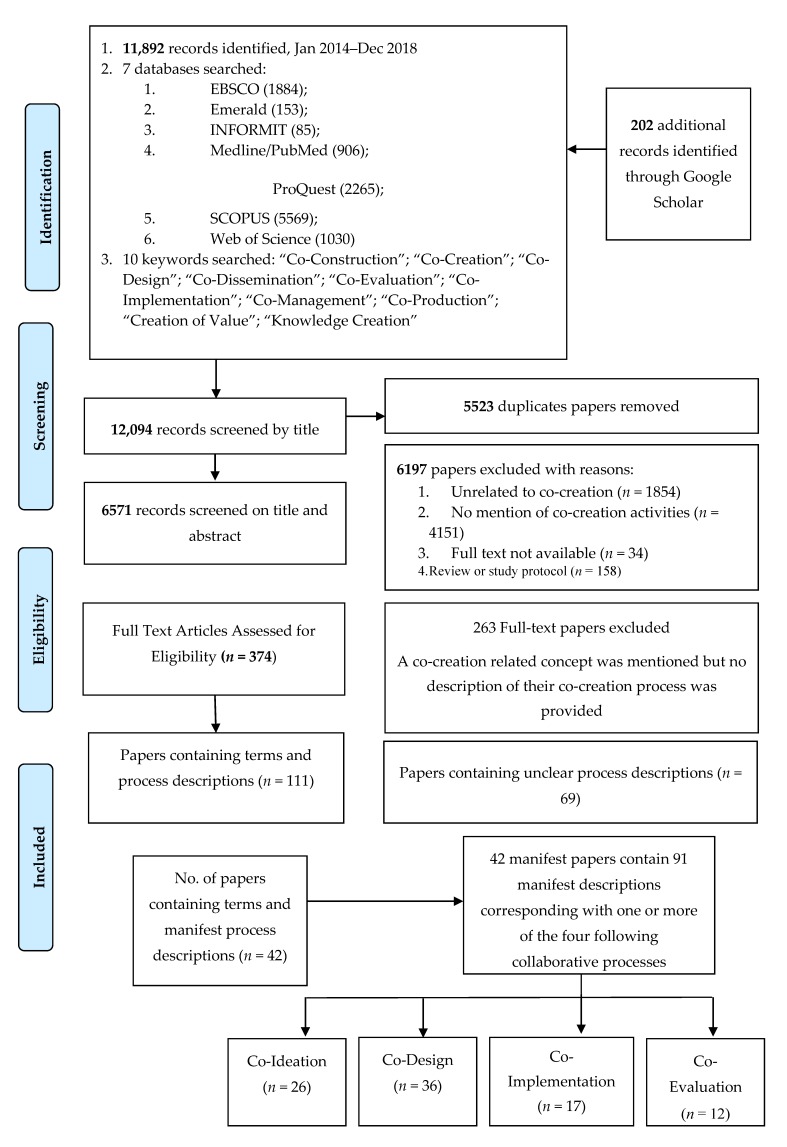
Preferred Reporting Items for Systematic Reviews and Meta-Analyses (PRISMA).

**Table 1 ijerph-17-02229-t001:** Co-creation-related keywords.

**1.1**	**Terms Identified during Snowball Search of Google Scholar**	**No. of Terms**
“co-creation (would also retrieve “value co-creation”, “resonant co-creation” and “co-creation of knowledge”), “co-assessment”, “co-commissioning”, “co-conception”, “co-construction”, “co-delivery”, “co-design”, “co-development”, “co-dissemination”, “co-evaluation”, “co-ideation”, “co-implementation”, “co-initiation”, “co-innovation”, “co-learning”, “co-management”, “co-planning”, “co-possibility”, “co-production”, “co-testing”, “knowledge creation” and “knowledge co-production”		22
**1.2**	**Most Frequently Used Keywords Appearing in Electronic Databases (>50 Records Retrieved)**	**No. of Terms**
“co-construction” OR “co-creation” OR “co-design” OR “co-dissemination” OR “co-evaluation” OR “co-implementation” OR “co-management” OR “co-production” OR “creation of value/value creation” OR “knowledge creation”		10

**Table 2 ijerph-17-02229-t002:** Use of co-creation-related terms in papers with a manifest description of their co-creation process, published in 2014–2018 (n = 42 papers).

Range of Terms	Industry				Papers	Total Number of Manifest Descriptions
	Community Based	Business & Marketing	Health & Welfare	Public Policy		
**Co-Ideation (n = 9 terms)**						**n = 26**
Co-Ideation		✓[41]			Agrawal & Rahman [41]	1
Co-Commissioning				✓[42]	Podgórniak-Krzykacz [42]	1
Co-Creation	✓[43,44]	✓[45,46]	✓[47]		Bryan-Kinns [44]; Jamin [47]; Wang [43]; Zhang [45]; Lapolla & Sanders [46]	5
Co-Design	✓[48,49,50,51,52]	✓[52,53]	✓[49,54]	✓[55]	Broadley & Smith [48]; Hjelmfors [49]; Ward [54]; De Jans [56]; Durl [53]; Westhorp [50]; Lam & Dearden [55]; Taffe [52]; Kwon [51]	9
Co-Development		✓[57]			Candi [57]	1
Co-Learning			✓[58]		Sharma & Conduit [58]	1
Co-Planning				✓[59,60]	Barbera [59]; Sicilia [60]	2
Co-Production	✓[61]		✓[62,63,64]	✓[65]	Dunn [62]; Hawkins [63]; Strokosch & Osborne [61]; Van Damme [64]; Thijssen & Van Doreen [65]	5
Co-Reflection		✓[66]			Mostafa [66]	1
**Co-Design (n = 7 terms)**						**n = 36**
Co-Design	✓[48,50,51]	✓[41,56,52,53,66,67]	✓[49,54,68]	✓[60,69,70,71]	Broadley & Smith [48]; Hetrick [68]; Hjelmfors [49]; Ward [54]; De Jans [56]; Durl [53]; Agrawal & Rahman [41]; Bessant [69]; Hahn-Goldberg [71]; Mostafa [66]; Sicilia [60]; Teichmann [67]; Westhorp [50]; Merickova [70]; Taffe [52]; Kwon [51]	16
Co-Development		✓[57,72]			Candi [57]; Cui & Wu [72]	2
Co-Creation	✓[44]	✓[45,46,73,74]	✓[47]		Bryan-Kinns [44]; Jamin [47]; Tommasetti [73]; Oyner & Korelina [74]; Zhang [45]; Lapolla & Sanders [46]	6
Co-Innovation		✓[75]			Wang [75]	1
Co-Production	✓[61,76]		✓[58,62,63,64,77,78]	✓[42]	Hawkins [63]; Dunn [62]; Sharma & Conduit [58]; Strokosch & Osborne [61]; Tu [76]; Van Damme [64]; Vennik [77]; Podgórniak-Krzykacz [42]; Wherton [78]	9
Knowledge Co-Creation		✓[79]			Tremblay & Jayme [79]	1
Value Co-Creation			✓[80]		Janamian [80]	1
**Co-Implementation (n = 5 terms)**						**n = 17**
Co-Implementation				✓[42,70,76]	Tu [76]; Podgórniak-Krzykacz [42]; Merickova [70]	3
Co-Creation	✓[43,44]	✓[46]	✓[58]		Bryan-Kinns [44]; Sharma & Conduit [58]; Wang [43]; Lapolla & Sanders [46]	4
Co-Delivery			✓[81]	✓[59,60]	Lwembe [81]; Barbera [59]; Sicilia [60]	3
Co-Design	✓[55,50]	✓[73]			Tommasetti [73]; Westhorp [50]; Lam & Dearden [55]	3
Co-Planning	✓[65]				Thijssen & Van Dooren [65]	1
Co-Production	✓[61,62]				Dunn [62]; Strokosch & Osborne [61]	2
Knowledge Creation		✓[79]			Tremblay & Jayne [79]	1
**Co-Evaluation (n = 7 terms)**						**n = 12**
Co-Creation			✓[47]		Jamin [47]	1
Co-Design	✓[56,50]				De Jans [56]; Westhorp [50]	2
Co-Evaluation		✓[41]		✓[60]	Sicilia [60]; Agrawal & Rahman [41]	2
Co-Management		✓[82]			Anderl [82]	1
Co-Production	✓[61]		✓[63,64]	✓[42]	Hawkins [63]; Strokosch & Osborne [61]; Van Damme [64]; Podgórniak-Krzykacz [42]	4
Co-Recovery		✓[66]			Mostafa [66]	1
Knowledge Co-Creation		✓[79]			Tremblay & Jayme [79]	1
**Total terms used/industry**	23	28	23	17		**n = 91**

**Table 3 ijerph-17-02229-t003:** Examples of latent and manifest content used to inform a new definition of “Co-Creation of Knowledge”.

Categories	Sub-Categories	Examples from the Literature
*Activity*	*Collaboration represented by the prefix “co” implying partnership and equality*	Union/collaboration of stakeholders (researchers and non-researchers) [83,84];“mutual knowledge exchange” [44]; “common understanding” [66]; shared vision and decision-making; meaningful engagement with participants [47,62]; equal voice and a collective vision [54,60]
*Activity*	*Ideation*	“generate” [18,83], “explore” [85]; “brainstorm ideas” [49], “provoke discussion” [44,78]; reflect on how to meet community needs, solve problems and improve service delivery [58,61,62,86,87]
*Activity*	*Designing*	taking of ideas (generated in the co-ideation phase); planning of the production of concrete outcomes (products, services or programs) [41,66]
*Activity*	*Implementing*	participation of stakeholders in the delivery of services and programs [42]
*Activity*	*Evaluating*	usability testing of prototypes and products [47] and state actors and the public to assess the quality of public services [70,86]; provide constructive feedback on services, interventions or products to researchers [56]; feedback may be collected through the administration of pre- and post-initiative questionnaires or engagement in focus groups [53,59]; feedback is collected and used to improve services [56]

**Table 4 ijerph-17-02229-t004:** Co-Creation of New Knowledge—Terminology and Operational Definitions.

Core Principles	Definition	Operationalising Examples
1. Rigorous research methods	Experimental or quasi experimental Evidence-based measures	- * RCT, SWD, MBD, WHO-QoL
2. Embedded	Integration of best-evidence measures into routine data collection processes [87] Programs or policies implemented into routine practice	- Data collected by agencies using evidence-based questionnaires and scales during delivery of programs, services or treatments
3. Contains Four Processes		
3.i Co-ideation	Engaging in open dialogue to share new and creative ideas for the solving of problems relating to new products, services, policies and programs	- Creativity Workshops - Regular meetings - Steering groups - Review statistics or records, service priorities or clinical guidelines - Review existing practices - Review existing research - Client/patient input
3.ii Co-design	Providing a description of the technical details of new products, services, procedures, policies or programs (prototype), as well as the research methods to be used (protocols). This process may include assessment of funding sources, availability of resources, research processes (e.g., ethics) and timelines.	- Convene staff meetings or workshops to document ideas and processes - Sessions to understand different roles/responsibilities and how change can be integrated into routine delivery of services - Client/patient input
3. iii Co-implementation	Implementing the co-designed program, policy or clinical procedures in accordance with the research protocol. This process may be a one-time collaborative event or an arrangement over the longer term.	- Staff training and development - Provide treatment/policy protocols - Identify barriers to, and enablers for, change - Execute procedures for data collection and analysis - Regular meetings for ongoing review of the process
3.iv Co-evaluation	Embedding data collection or other formal research techniques into the co-implementation process. Researchers with relevant bio-statistical skills undertake analyses. Co-interpretation of the meaning and implications of the results.	- Modify existing or implement new data collection systems - Ongoing staff training and development in the importance/processes for using the data collection system - Workshops to consider/interpret the findings. - Client/patient input

* RCT: Randomised Controlled Trial; SWD: Step Wedge Design; MBD: Multiple Base Design; WHO-QoL:World Health Organization Quality of Life.

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
