# Peer review of "What is the Co-Creation of New Knowledge? A Content Analysis and Proposed Definition for Health Interventions"

_ijerph, 2020, doi:10.3390/ijerph17072229_

Round 1
Reviewer 1 Report
Dear Authors,
despite my previous reject suggestion, I found your paper improved.
Author Response
Thank you for your support. We appreciate your insight that has contributed to improving the quality of the paper and we value the time and effort you have invested in providing feedback.
Reviewer 2 Report
Dear Authors
Thank you for making revisions based on previous feedback. I can see the changes made across the document. Sections 3.2, Table 3 show marked improvement. I still think you need to strengthen the conclusions and draw clear theoretical and practitioner novelty (as a result of the paper).
best of luck.
Author Response
Thank you for your continued support of our paper.
We have added additional information into the conclusion section to cover the theoretical and practitioner novelty value. L383-385 and L389-394.
We have also conducted another round of checking for spelling and grammatical errors and believe we have corrected all of these mistakes.
Reviewer 3 Report
Changes made to the article have helped improve its quality. Highlight
the greater precision on the object of analysis both in the title and in the
work introduction. The justifications related to the methodology and the results
have been very useful.
Author Response
We appreciate your insight that has contributed to improving the quality of the paper and we value the time and effort you have invested in sharing your valuable feedback.
This manuscript is a resubmission of an earlier submission. The following is a list of the peer review reports and author responses from that submission.
Round 1
Reviewer 1 Report
Dear Authors,
I suggest rejection for your work. In my opinion, it has limited value for scholars. In addition, I found it confused, as it is not clear if the health sector is important or not for the analysis. The output is more or less a definition.
Another major concern refers to the 69 papers excluded because they were too ambiguous.
I hope my comments will not discourage your research effort.
Reviewer 2 Report
Thank you for this paper. the topic is interesting, and I really think the authors should give it a stronger backbone. Here are my comments to the authors, from what I understood from reading the paper.
Line 54 did not include insufficient detail (delete "in")
Line 55 take into account (add word)
What the authors mean by translational research needs to be clarified: research which is useful for practice? Research which is usable by practitioners? Research which is understandable by the layman? Do the authors wish to discuss this from the point of view of epistemology (constructivism vs positivism?). I was surprised not to see a critical reflection on how action-research may contribute to the argument made by the author (point 3 in the introduction).
Also which field of research does this study take root in? in the field of health promotion research for eg, the argument would have to be different….
The authors could (or should) differentiate between translational research / translational models / translated data / translatable data..
What reference supports the “co-creation of new knowledge” framework?
Co-creation of knowledge is not the same as co-creation of an intervention / or a research project.
The focus of this paper could be clearer. Is it about co-creation of programmes and how research findings can contribute to them? Is it about action research? Is it about the co-creation of knowledge, not necassily used to design a programme?
Why the choice of 2014-2018 for the litt search? (presented in the aims, otherwise they would have to delete this)
It would be useful to formulate research questions. 3 aims are presented: why such aims?
Line 124 I would change “IS a method..” to “can be.. “ as there are more than one way to carry out content analysis.
How did the authors identify “latent” descriptions? What framework / definition was used? The authors claim that there is no definition of cocreation…How many researchers screened the articles?
What are the inclusion criteria?
I would make the results and contribution of this paper clearer. How do authors contribute to research with their paper? And in what field? To which question do they wish to answer?
I cannot see how the authors concluded with their definition from the results they present. The process of data analysis would have to be more explicitly presented.
Conclusions in line 50 are not convincing..line 55, conclusions seem to result from a circular argument. Authors looked for the 4 processes and found the same processes as a result. Also, how can they argue that co-creation comprises four collaborative processes? What is the theoretical backbone to this assumption?
Line 86, the authors say they have provided the reader with a definition of co-creation, when the say earlier in the manuscript that they have provided a definition of the cocreation of new knowledge.. which are two different concepts. The authors really have to specify their focus more.
I hope my comments are useful. And I wish to emphasize that the topic is highly relevant and timely. I would make sure this article makes a good case for the co-creation of knowledge (between whom and whom?)
Reviewer 3 Report
Dear Authors
Thank you for the paper. It is a topic of great professional and academic relevance.
Some comments to help improve the paper.
Abstract
although the elements of co-creation are unpicked, what is new knowledge is missing!Keywords need to be re-examined as new knowledge does not appear.
Introduction
need to introduce knowledge-based economy, importance of innovation (new knowledge) in the current global context 3 translation models are listed but not elaborated - why are these models important or why are they chosen and included? page 2, line 80 paragraph on definitions and perspectives of co-creation needs further elaboration - it is the main part of the paper, split the paragraph based on key themes from literature need to add a discussion on what is new knowledge? need to unpack co-creation in the services context (see Prahalad, C.K. and Ramaswamy, V., 2004. Co-creation experiences: The next practice in value creation. Journal of interactive marketing, 18(3), pp.5-14.) provide definitions from literature for the 10 most cited co-creation wordsMaterial and method
This section explains the process quite clearly. have two key concerns:
42 papers is a small sample size given that 6,571 papers were reviewed papers on knowledge co-creation/value co-creation need further attention (add paper Ashok, M., Narula, R. and Martinez-Noya, A., 2016. How do collaboration and investments in knowledge management affect process innovation in services?. Journal of Knowledge Management, 20(5), pp.1004-1024.) elaborate the process involved in taking decision to exclude papers - how did multiple authors contribute to this process? was it robust?Reporting and Results Phase
good to see trustworthiness of the coding.
research context plays a key role in co-creation, thus this aspect needs to be further explored-explained in the results table 4 - do the operational examples map to the definitions provided? if yes they need to be clearly marked which examples map to which definitionsDiscussion
could elaborate on the implications for theory (as a sub-section)Conclusions
this section is really short, needs to be beefed up to highlight novely of this paper and why it has professional and academic relevance (to improve chances for citation)best of luck.
Reviewer 4 Report
The paper analyzes a relatively new concept in different disciplines. It contributes positively to the generation of debate on the co-creation of new knowledge. Likewise, it helps to establish a broad definition that includes all the peculiarities of the concept.
As for the structure of the paper, the objectives of the investigation are clear in its introduction. In the materials and methods part, being a qualitative analysis, it would be advisable to introduce some improvements:
a) Explain in more detail the sources and databases consulted, as well as justify the temporary choice of the articles consulted.
b) Figure 1 is very clarifying. The explanation should specify more the sequence of organization of the articles and the criteria used to not select some of them.
c) Explain in detail the information in tables 2, 3 and 4. The tables serve as support, but to guide the reader should explain the elements that make up these tables and how they are related.
d) In the section related to the integrity of the model, detail the statistical contrasts used.
The discussion of the results is detailed. A table could be made with the comments in the discussion section that contribute to the configuration of the concept of co-creation of knowledge.
Finally, the article highlights the limitations with which it is found, which provides consistency to the analysis performed.